# Validity and Applicability of the Global Leadership Initiative on Malnutrition (GLIM) Criteria in Patients Hospitalized for Acute Medical Conditions

**DOI:** 10.3390/nu15184012

**Published:** 2023-09-16

**Authors:** Laia Fontane, Maria Helena Reig, Sonika Garcia-Ribera, Miriam Herranz, Mar Miracle, Juan Jose Chillaron, Araceli Estepa, Silvia Toro, Silvia Ballesta, Humberto Navarro, Gemma Llaurado, Juan Pedro-Botet, David Benaiges

**Affiliations:** 1Department of Endocrinology and Nutrition, Consorci Sanitari Alt Penedès-Garraf, Espirall, 61, 08720 Vilafranca del Penedès, Spain; lfontane@csapg.cat (L.F.); hreig@csapg.cat (M.H.R.); sgarciari@csapg.cat (S.G.-R.); mherranz@csapg.cat (M.H.); jjchillaron@csapg.cat (J.J.C.); aestepa@csapg.cat (A.E.); storo@csapg.cat (S.T.); sballesta@csapg.cat (S.B.); hnavarro@csapg.cat (H.N.); 2Department of Endocrinology and Nutrition, Hospital del Mar, Passeig Marítim, 25-29, 08003 Barcelona, Spain; gllaurado@psmar.cat (G.L.); jpedrobotet@psmar.cat (J.P.-B.); 3Facultat de Farmàcia i Ciències de l’Alimentació, Universitat de Barcelona, Joan XXIII, 08028 Barcelona, Spain; marmiracle98@gmail.com; 4Department of Medicine, Universitat Pompeu Fabra, Plaça de la Mercè, 10-12, 08002 Barcelona, Spain; 5Institut Hospital del Mar d’Investigacions Mèdiques (IMIM), Dr. Aiguader, 80, 08003 Barcelona, Spain; 6Department of Medicine, Universitat Autònoma de Barcelona, Campus Universitari Mar., Dr. Aiguader, 80, 08003 Barcelona, Spain; 7Center for Biomedical Research on Diabetes and Associated Metabolic Diseases (CIBERDEM), Instituto de Salud Carlos III (ISCIII), 28029 Barcelona, Spain; 8Centro de Investigación Biomédica en Red de la Fisiopatología de la Obesidad y Nutrición, Monforte de Lemos Avenue, 3-5, Pavilion 11, Floor 0, 28029 Madrid, Spain

**Keywords:** global leadership initiative on malnutrition, nutrition assessment, malnutrition, hospitalized patients, length of stay, applicability

## Abstract

(1) Background: The objectives of this study were to evaluate the concurrent and predictive validity and the applicability of the global leadership initiative on malnutrition (GLIM) criteria in patients hospitalized for acute medical conditions. (2) Methods: prospective cohort study with patients hospitalized for acute medical conditions. For validation, the methodology proposed by the GLIM group of experts was used. Sensitivity and specificity values greater than 80% with respect to those for the subjective global assessment (SGA) were necessary for concurrent validation. The time necessary to complete each nutritional assessment test was determined. (3) Results: A total of 119 patients were evaluated. The SGA was applied to the entire cohort, but the GLIM criteria could not be applied to 3.4% of the patients. The sensitivity and specificity of the GLIM criteria with respect to those for the SGA to detect malnutrition were 78.0 and 86.2%, respectively. The GLIM predictive validity criterion was fulfilled because patients with malnutrition more frequently had a hospital stay >10 days (odds ratio of 2.98 (1.21–7.60)). The GLIM criteria required significantly more time for completion than did the SGA (*p* = 0.006). (4) Conclusion: The results of this study do not support the use of the GLIM criteria over the SGA for the diagnosis of malnutrition in patients hospitalized for acute medical conditions.

## 1. Introduction

Malnutrition related to disease is a problem especially present in the hospital environment and is associated with an increase in morbidity, mortality, hospital stay, and, consequently, health and social costs [1]. Additionally, the provision of nutritional support to malnourished patients, in addition to improving aspects related to nutritional status, is also related to improvements in health parameters such as hospital stay and mortality [2]. For this reason, it is vitally important to detect and treat hospital malnutrition.

The prevalence of malnutrition in hospitalized patients is highly variable, ranging from 12% to 87% [3,4,5]. The high variability in the rates of malnutrition in different studies is due both to the different profiles of patients analyzed and to the methods used for the diagnosis of malnutrition. To date, there is no consensus on the best nutritional assessment test for the diagnosis of malnutrition. The most widely used test in the general population and specifically in hospitalized patients is the subjective global assessment (SGA). The SGA is a widely validated nutritional assessment tool used in clinical practice because it is simple, applicable to all patients, noninvasive, inexpensive, fast, and can be performed at the bedside. Although the subjective nature of the SGA could be a limitation, it has a good interpersonal correlation if administered by trained personnel [6].

In 2018, the global leadership initiative on malnutrition (GLIM) collaborated with most of the scientific societies related to nutrition in an attempt to standardize the diagnosis of malnutrition in clinical settings, proposing the GLIM criteria as a standardized method for the diagnosis of malnutrition. These criteria have two main potential advantages: they minimize subjectivity in the diagnosis and stratify malnutrition by degree [7]. These criteria are currently limited as a nutritional assessment method because they are not widely validated in different settings and populations; the creators of the consensus have encouraged the scientific community to validate the criteria and to do so following strict methodology [8]. There are very few studies that have tested the validity of the GLIM criteria in hospitalized patients, and some studies have validated the criteria [9,10,11] and others have not [12,13,14,15,16], especially with regard to concurrent validity; furthermore, only a select few are prospective studies [10,13,14,16]. Given this controversy, more validation studies, ideally prospective studies that assess both concurrent and predictive validity, are needed in hospitalized patients.

In addition to the need to broadly validate these criteria, it is important to take into account two possible limitations of the GLIM criteria that may result in a loss of applicability in daily clinical practice for hospitalized patients. First, the GLIM criteria, in addition to requiring responses by patients to nutrition-related questions, including weight history, changes in intake, and gastrointestinal symptoms, require the assessment of the loss of muscle mass through objective data and of the degree of systemic inflammation. Thus, the time for completing the GLIM criteria is longer than that for completing other nutritional assessment tools, such as the SGA. This is highly relevant because, currently, nutrition departments are under-resourced, especially regarding highly qualified personnel dedicated to nutritional support [17,18]. Second, previous validation studies have excluded patients who cannot stand upright or who have edema, situations that are not uncommon during the first days of admission [10,11,13]. Therefore, patients for whom the assessment of the phenotypic criteria could not be carried out completely have been taken into account. Taking into account the importance of detecting malnutrition in hospitalized patients, it is highly relevant for nutritional assessment tests to be able to be applied to any type of patient. The validity of the GLIM criteria could be lower for these patients, for whom the entire test cannot be fully applied.

Considering these aspects, the main objective of this study was to determine the concurrent and predictive validity of the GLIM criteria in hospitalized patients with acute medical conditions. The secondary objectives were to compare the time required to complete the GLIM with that required to complete the SGA and to determine the percentage of patients for whom not all phenotypic criteria could be measured and the concurrent validity to detect malnutrition.

## 2. Materials and Methods

### 2.1. Study Design and Study Population

This was a prospective cohort study conducted with patients hospitalized for acute medical conditions at Corsorci Sanitari de l’Alt Penedès-Garraf between April and October 2022. The patients included in the study were older than 18 years, hospitalized in medical units for an acute medical problem, and had to be conscious and oriented in time and space. As exclusion criteria, the study did not include patients with nonacute pathology, critical emergencies, or scheduled admissions; patients for whom admission was expected to be less than 48 h, patients hospitalized in surgery or traumatology departments; pregnant women; and patients who did not give or were not able to give their informed consent. Patients with active SARS-CoV-2 infections were also excluded due to the difficulties of performing nutritional assessments in isolation.

Patient evaluations and data collection were carried out by qualified nutritionists (dietitian-nutritionists) during the first 72 h after hospital admission. A nutritional assessment was performed using the SGA and the GLIM criteria in three blocks. The first block included items that were common to both the SGA and the GLIM criteria. The second block was dedicated to questions unique to the SGA. The SGA questions were posed first because of their subjective nature; if the GLIM criteria had been assessed first, the results could have influenced the response to the SGA questions. After the second block, the patients were classified into correct nutritional status (A), risk of malnutrition or moderate malnutrition (B), or severe malnutrition (C) based on a subjective evaluation of the different items assessed. The third block focused on aspects unique to the GLIM criteria (Appendix A).

The definition of each phenotypic and etiological criterion of the GLIM criteria is shown in Table 1. A patient was considered malnourished if he or she had one phenotypic criterion and one etiological criterion [7]. The degree of malnutrition was assessed by weight loss and low BMI.

For patients who could not stand and therefore could not be weighed, the weight loss criterion could not be determined. In these cases, BMI was assumed to be unaltered in patients in whom the diagnosis was established or it was clear from the physical examination that they were overweight or obese. In case of doubt, the BMI criterion was considered not measurable. For patients with edema or amputation in the lower extremities, muscle mass was considered not measurable. If one or two phenotypic criteria could not be determined, the available phenotypic criteria were used for the assessment. In the event that none of the three phenotypic criteria could be determined, the GLIM assessment could not be performed.

Each of the blocks was timed separately. The total time to complete the SGA was calculated by adding the times for the common block and the specific SGA block, and the total time to complete the GLIM criteria was calculated by adding the times for the common block and the specific GLIM block.

The following variables were also collected: sociodemographic, medical, and analytical data; the Charlson comorbidity index [20], the Barthel index, and days of hospital stay.

### 2.2. Statistical Analysis

Accepting an alpha risk of 0.05 and a beta risk of 0.2 in bilateral contrasts, 117 patients were required to detect a difference equal to or greater than three days of hospital stay between subjects with and without malnutrition. A malnutrition rate of 29.7% was assumed for the GLIM criteria based on a study carried out in hospitalized patients in Spain [21], a common standard deviation of five was applied based on data from our center in 2021, and the estimated loss rate was 10%.

A descriptive analysis of the set of variables collected was carried out. Continuous variables with a normal distribution are expressed as the mean and the standard deviation for the total number of patients; if the distribution is not normal, continuous variables are expressed as the median and interquartile range for the total number of patients. The qualitative variables are expressed as relative and absolute frequency distributions. The normality of the variables was evaluated using quantile graphs (qqplot) and the Shapiro–Wilks test.

Comparisons between the groups (patients with vs. without malnutrition and patients in whom one or two phenotypic criteria could not be performed vs. patients with complete nutritional assessments) were conducted using the Student’s t test; the chi-square test was used for categorical variables. For those variables that did not follow a normal distribution, a logarithmic transformation was performed. For the analysis, subjects with moderate and severe malnutrition were grouped.

For the study of concurrent validity, the methodology proposed by the GLIM group of experts was used [8]. ROC (receiver operating characteristic) curves and sensitivity, specificity, and positive and negative predictive values for the GLIM criteria were calculated using the SGA as the reference method. The patients in whom the three phenotypic criteria could not be determined were excluded from the analysis of predictive sensitivity. Sensitivity and specificity values greater than 80% were considered necessary to confirm concurrent validity. Concurrent validity was also analyzed, excluding the subgroup of patients in whom one or two phenotypic criteria could not be assessed. 

For the analysis of predictive validity, the group of experts defined an OR ≥ 2 for a categorical variable as necessary to confirm predictive validity. Thus, hospital stays were dichotomized into >10 days and ≤10 days, and logistic regression analysis adjusted for age, sex, the Barthel index, and the Charlson comorbidity index was used. This cut-off point was selected in accordance with previous studies [10,22]. The analysis was performed for both the GLIM criteria and the SGA. Finally, the time required to complete the SGA was compared with the time required to complete the GLIM criteria. The analyses were carried out using the statistical package R version 4.1.0 (2021-05-1) for Windows. A value of *p* < 0.05 was considered significant.

## 3. Results

### 3.1. General Characteristics of the Sample

A total of 119 patients were included during the study period. Of these, 50.4% were women, and 97.5% were Caucasians. The mean age of the patients was 65.2 ± 14.9 years. The main reasons for hospitalization were infectious disease (31, 26.1%), lung disease (21, 17.7%), heart disease (11, 9.2%), digestive disease (14, 11.8%), neurological disease (7, 5.9%), and other causes (35, 29.4%).

All patients were assessed by the SGA; 52 patients (43.7%) met the criteria for malnutrition: 32 (26.9%) had moderate malnutrition or a risk of malnutrition, and 20 (16.8%) had severe malnutrition. For 4 patients (3.4%), none of the 3 phenotypic criteria could be assessed; therefore, the GLIM criteria were not assessable. The remaining 48 (40.3%) met the criteria for malnutrition: 23 (19.3%) had moderate malnutrition, and 16 (13.4%) had severe malnutrition; for 9 (7.6%), the degree could not be determined. Inflammation was assessed using the CRP value for all but four patients, for whom inflammation was determined using clinical criteria (one met the criteria for malnutrition according to the GLIM criteria, but the rest did not). Table 2 shows the characteristics of the patients with and without malnutrition as determined using the SGA and the GLIM criteria. For the patients with malnutrition as determined by both the GLIM criteria and the SGA, there was a predominance of males, Charlson comorbidity index scores were higher, and CRP levels were higher. In addition, they presented worse nutritional parameter values for albumin, BMI, weight loss, and calf circumference. Patients with malnutrition as determined by the SGA had a lower Barthel index score, a difference that was not observed with the GLIM criteria.

### 3.2. Concurrent Validity

Nine of the 65 patients (13.8%) without malnutrition, as determined using the SGA, were classified as malnourished using the GLIM criteria (false-positives), and 11 patients of the 50 (22.0%) with malnutrition, as determined using the SGA, were not malnourished based on the GLIM criteria (false-negatives). The concurrent validation criteria were not met because the sensitivity was less than 80% (Table 3).

Of the patients, 20 presented severe malnutrition according to the SGA, and 16 (80%) also presented severe malnutrition according to the GLIM criteria. Of the remaining patients, 3 (15%) were diagnosed as moderately malnourished according to the GLIM criteria, and 1 (5%) was determined to have normal nutrition. Of the 32 patients with moderate malnutrition according to the SGA, 19 (59.4%) were in the same category according to the GLIM criteria; for 10 (31.3%) of the remaining patients, they were determined to have normal nutrition using the GLIM criteria; for 1 (3.1%), the degree of malnutrition was not assessable; and for 2 (6.3%), the GLIM criteria were not applicable. Regarding the 67 patients without malnutrition, as determined using the SGA, 56 (83.7%) were diagnosed as not having malnutrition by the GLIM criteria, and 9 (13.4%) presented malnutrition (1 moderate and 8 without grades); for two of the patients, the GLIM was not applicable (3.1%).

Regarding the nine patients in whom the degree of malnutrition could not be determined using the GLIM criteria, eight did not present malnutrition according to the SGA, and only one presented moderate malnutrition. In fact, eight of the nine false-positives in the entire cohort had no or moderate malnutrition. These are patients in whom the criteria of low BMI and weight loss were negative or not assessable and who did present reduced MM.

In the detailed analysis of the 11 patients who were false-negatives, all met some phenotypic criteria but none of the two etiological criteria. Regarding the phenotypic criteria of these patients, all had positive weight loss, 2 had a low BMI, and 5 had a low MM. The inflammation criterion was negative for presenting CRP < 5 mg/dl.

### 3.3. Predictive Validity

As seen in Table 4, a higher percentage of patients with malnutrition, as determined by both the GLIM criteria and the SGA, had a hospital stay of more than 10 days. In the multivariate analysis, the presence of malnutrition, as determined by the GLIM criteria, increased the risk of having a hospitalization greater than 10 days by 2.98 times (*p* = 0.019), thus fulfilling the predictive validation criterion. The odds ratio (OR) adjusted for the SGA was 6.6 (*p* < 0.001). 

Seven patients died during hospitalization, with four of them having a hospital stay of less than 10 days. After excluding hospital mortality, the predictive validity of GLIM and VSG for detecting hospital stays longer than 10 days remained significant (model 2).

### 3.4. GLIM Applicability

In addition to the four patients (3.4%) for whom none of the three phenotypic criteria could be assessed, for 20 patients, only one or two phenotypic criteria could be assessed (16.8%). Of these, only one phenotypic criterion could be assessed in 13 patients and two criteria in seven patients.

The criterion for which the most values were missing was BMI, which could not be determined in 20 patients (16.8%), followed by weight loss, which could not be assessed in 18 patients (15.1%), and loss of muscle mass (MM), which could not be determined in 7 patients (5.9%). The patients with 1 or 2 nonassessable phenotypic criteria were differentiated from those who could be assessed by older age and a lower Barthel index (Table 5).

The predictive validity test, in which patients with one or two nonassessable criteria were excluded, also failed to meet the validation criteria (Table 3). Finally, as seen in Table 3, when excluding the nine patients for whom the degree of malnutrition could not be determined through the GLIM criteria, the specificity improved but the sensitivity remained below 80%.

### 3.5. GLIM Time

The time spent completing the GLIM criteria was 5′ 9″ ± 1′ 58″, longer than that required for the SGA (4′ 27″ ± 2′ 14″) (*p* = 0.006).

## 4. Discussion

This study provides new evidence on the validity of the GLIM criteria for patients hospitalized for acute medical conditions. In addition, it provides data on the applicability of these criteria, which can be very useful in clinical practice.

First, one of the most remarkable results of this study is that the concurrent validation standards proposed by the experts who designed the GLIM criteria were not met (4). Thus, although the specificity to detect malnutrition was greater than 80%, the sensitivity was 78.0%. These results have an impact on current clinical practice because they suggest that the GLIM criteria should not be used for patients hospitalized for acute medical conditions. Previous studies on hospitalized patients reported heterogeneous results, with some reporting that the GLIM criteria are valid [9,10,11] and some reporting that they are not [12,13,14,15,16]. The differences can be explained by different study designs, e.g., prospective or retrospective studies, and differences in the typology of hospitalized patients (medical, medical-surgical, or geriatric), the exclusion criteria, or the way to determine each criterion of the GLIM criteria.

A study most similar to the one reported herein was conducted by Brito JE. et al. [10]. In a prospective study with 601 hospitalized medical patients, the GLIM criteria were compared with the SGA, and the GLIM criteria were validated, obtaining sensitivity and specificity values of 87 and 82%, respectively. Unlike the study by Brito JE et al., validation was not achieved in the present study, likely because of the high percentage of false-negatives that caused the sensitivity to be less than 80%. In a detailed analysis of the 11 false-negatives, these patients met some phenotypic criteria but did not meet either of the two etiological criteria. Comparing the results reported by Brito JE. et al. [10] with the findings of this study and focusing on the etiological criteria, the prevalence of alterations in dietary intake was similar, but the presence of inflammation in the patients of this study (44.3%) was well below that reported by Brito JE. et al. (83.4%). In the study design, we chose CRP to define inflammation for two reasons: 1) CRP is one of the measures proposed by the authors of the GLIM criteria to define inflammation, and 2) the measurement of CRP would reduce subjectivity, following the approach of another study that validated the GLIM criteria [9]. The results of the present study seem to indicate that CRP should not be used as the sole criterion to define inflammation in hospitalized patients with acute medical conditions. It is not logical that in this profile of patients, less than half met the criteria for inflammation, also affecting the validity of the criteria. Thus, it could be more appropriate to define inflammation through admission diagnoses, combine CRP with another indicator, or define new CRP cutoff points.

The second most relevant aspect of this study is that the predictive validity criteria were met, with an OR of 2.98, demonstrating that patients with malnutrition according to the GLIM criteria more frequently had a hospital stay greater than 10 days. In addition, this relationship remained significant in the multivariate analysis, taking into account variables classically related to hospital stay, such as age, sex, the Charlson index, and the Barthel index. In this sense, most studies support the predictive validity of the GLIM criteria [10,11,14,22,23,24], with studies that, like the present one, have demonstrated an association with hospital stay [22]. In a study by Wang P. et al. [22], a group of hospitalized patients pending esophagectomy for esophageal cancer with malnutrition, as determined using the GLIM criteria, had a longer hospital stay, with an OR of 3.84 for moderate malnutrition and 7.38 for severe malnutrition. In addition, other studies have established that the GLIM criteria meet the predictive validity criteria for other health events, such as mortality [10,11,14,23,24]. Of the studies in which predictive validity was not achieved, Tan S. et al. [25] used hospital stay as a continuous variable rather than a categorical variable, as recommended. Brito JE. et al. [10] observed a significant relationship with hospital stay, but the validation criterion was not met because the OR was not greater than two.

Another notable aspect of predictive validity in the present study is that malnutrition measured by the SGA was also associated with a longer hospital stay. The predictive validity of the SGA has already been widely determined in previous studies [6]. Of the studies that, like the present one, have explored both the predictive validity of the GLIM criteria and the SGA, Balci C. et al. [11] conducted a retrospective analysis of hospitalized patients in which the GLIM criteria and the SGA presented a similar OR for mortality. In the present study, the SGA had a predictive power more than double that of the GLIM criteria, with an OR of 6.16. Consequently, the results of the present study suggest that the SGA has greater power in predicting a prolonged hospital stay than do the GLIM criteria.

In daily clinical practice, a tool for assessing nutritional status, in addition to being valid, must meet other qualities, such as applicability. It is not uncommon for patients to be unable to get out of bed upon admission or due to a medical prescription, nor is the presence of edema uncommon. In this clinical setting, the SGA is always applicable; in contrast, in our study, for 3.4% of the patients, the GLIM criteria could not be used because none of the three phenotypic criteria were available. Although this is a low percentage, with the use of the GLIM criteria, certain groups of patients cannot be classified as normal or malnourished, and there is a question as to whether or not nutritional therapy should be administered. Specifically, in the present study, two of the four patients with nonassessable GLIM criteria had malnutrition with SGA.

Another point to highlight regarding the applicability of GLIM is that for 16.8% of the patients analyzed with the GLIM criteria, one or two phenotypic criteria could not be assessed. The criteria with the most missing values were low BMI (16.8%) and weight loss (15.1%), because we did not have weight or height to calculate these two parameters. These patients with one or two nonassessable phenotypic criteria presented a lower Barthel index, suggesting that the most dependent patients are those for whom the GLIM criteria cannot be fully assessed. According to our initial hypothesis, by excluding these patients, as most studies have done, the concurrent validity should have improved. However, this hypothesis was confirmed because the false-positives decreased but not the false-negatives.

Another aspect of the applicability of the GLIM criteria that should be addressed is that for nine patients, a diagnosis of malnutrition was made through the GLIM criteria, but it was not possible to determine the degree of malnutrition. For these patients, the phenotypic criterion was positive only because of a reduction in MM through the determination of CC; therefore, malnutrition could not be further stratified into moderate or severe. This is a disadvantage when compared to the SGA, for which all patients were classified by different degrees of malnutrition. Furthermore, of the nine false positives present in the entire cohort, eight were in the group for whom the degree of malnutrition could not be determined. Thus, when they were excluded, the specificity improved to approximately 100%, but the sensitivity remained below 80%. This result highlights the difficulty of assessing muscle compartments in hospitalized patients, which in turn may affect the applicability of the GLIM criteria. Thus, the group of experts proposed that muscle mass could be assessed by validated techniques such as DEXA, CT, muscle ultrasound, impedance, or anthropometry. In this present study, as in that by Brito JE. et al. [10], CC was used because of the limitations of the other techniques with regard to hospitalized patients. Body composition techniques such as DEXA and CT are not applicable to all hospitalized patients due to time, cost, or exposure to radiation. Impedance measurements have decreased validity in hospitalized patients because of the conditions required for measurement, such as prior fasting, the absence of ascites or edema, non-transportable machinery, patient compliance for certain impedance measurements, and time [26]. In view of the results of this study, the usefulness of other anthropometric measures or the use of muscle ultrasound should be explored, although the latter requires time and trained personnel.

Finally, we determined the time invested in each test and found that the time to complete the GLIM criteria was longer. This factor is very relevant in routine clinical practice because of the lack of dietitians and nutritionists in our health system [17,18], which causes an overload of nutrition departments in different centers. Nevertheless, the actual impact of this time-saving will depend on the number of daily nutritional assessments conducted in each center, as the differences, while statistically significant, were just over 40 s per test.

The present study is not without limitations. First, although recommended by the group promoting the GLIM criteria [8], this study did not evaluate the reliability of or agreement between researchers. In view of the results of the study, this aspect is important to investigate, especially for the assessment of the muscle compartment. The second limitation is that the method used to determine muscle mass does not yet have cutoff points to differentiate moderate and severe malnutrition, a relevant fact that also affects other methods proposed by the authors such as muscle ultrasound or CT. Third, we did not evaluate predictive validity using other outcomes, such as mortality. Finally, this is a single-center study with a small sample size; therefore, the results cannot be extrapolated to other settings.

## 5. Conclusions

In conclusion, the results of this study do not support the use of the GLIM criteria over the SGA for the diagnosis of malnutrition in patients hospitalized for acute medical conditions for 3 reasons: it has a lower predictive validity than that for the SGA; it does not meet the concurrent validity criteria; and it is less applicable than the SGA. More studies are needed to establish the best method to define inflammation and efficient tools for the assessment of muscle mass in hospitalized patients.

## Figures and Tables

**Table 1 nutrients-15-04012-t001:** Malnutrition diagnosis using the GLIM criteria.

	Phenotypic Criteria			Etiological Criteria	
	Weight Loss (%)	Low BMI (kg/m^2^)	Reduced Muscle Mass	Reduced Dietary Intake (or Absorption)	Inflammation
Moderate malnutrition	5–10% in 6 months or 10–20% in more than 6 months	<20 in patients <70 years or <22 in patients ≥70 years	CC ≤ 33 cm in women and ≤34 cm in men, adjusted for BMI [19]	≤50% of the intake with respect to the usual in the last week or any reduction >2 weeks or the presence of diseases that alter the absorption of food	CRP > 5 mg/dL; when CRP was not available, the definition of metabolic demand based on the pathophysiology of the disease
Severe malnutrition	>10% in 6 months or >20% in more than 6 months	<18.5 in patients <70 years or <20 in patients ≥70 years			

Abbreviations: BMI = body mass index; CC = calf circumference; CRP = C-reactive protein.

**Table 2 nutrients-15-04012-t002:** Characteristics of patients with and without malnutrition as determined using the SGA and the GLIM criteria.

Variable	GLIM			SGA		
	Well Nourished (*n* = 67)	Malnourished (*n* = 48)	*p* Value	Well Nourished (*n* = 67)	Malnourished (*n* = 52)	*p* Value
Sociodemographic data						
Age (years)	64.8 ± 15.8	66.0 ± 14.1	0.332	64.2 ± 15.2	66.5 ± 14.4	0.204
Women (%)	61.2	35.4	0.006	61.2	36.5	0.008
Caucasian (%)	98.5	95.8	0.171	98.5	96.2	0.185
Medical data						
Charlson index	1.0 (0–3)	2.8 ± 2.6	<0.001	1.5 ± 1.7	2.9 ± 2.6	<0.001
Barthel index	83.7 ± 26.6	82.8 ± 22.0	0.429	86.3 ± 23.7	78.6 ± 25.5	0.046
Analytical parameters						
Albumin (mg/dL)	3.2 (2.9–3.5)	2.7 ± 0.6	<0.001	3.2 ± 0.5	2.7 ± 0.6	<0.001
CRP (mg/dL)	1.2 (0.5–3.9)	21.3 ± 31.1	0.013	8.8 ± 31.7	18.6 ± 30.5	0.048
Nutritional characteristics						
Current weight (kg)	74.2 ± 16.4	67.2 ± 13.3	0.012	75.4 ± 16.2	67.2 ± 14.6	0.004
BMI (kg/m^2^)	27.6 ± 5.9	24.3 ± 5.1	0.003	27.9 ± 5.8	24.9 ± 6.5	0.009
Weight loss (kg)	0.3 ± 4.5	6.9 ± 6.2	<0.001	-0.8 ± 4.1	7.8 ± 4.8	<0.001
CC (cm)	33.8 (31.5–36.0)	31.9 ± 2.9	0.007	33.8 ± 4.7	32.1 ± 3.1	0.013
Presence of etiological criteria						
Altered weight loss (*n*, %)	14 (20.9%)	34 (70.8%)	<0.001	4 (6%)	44 (84.6%)	<0.001
Nonmeasurable weight loss (*n*, %)	8 (11.9%)	6 (12.5%)		12 (17.9%)	6 (11.5%)	
Low BMI (*n*, %)	4 (6.0%)	10 (20.8%)	0.031	2 (3%)	12 (23.1%)	0.003
Nonassessable BMI (*n*, %)	8 (11.9%)	8 (16.7%)		13 (19.4%)	7 (13.5%)	
Reduced MM (*n*, %)	23 (34.3%)	30 (62.5%)	0.005	26 (38.8%)	27 (51.9%)	0.2
Nonassessable MM (*n*, %)	1 (1.5%)	2 (4.2%)		3 (4.5%)	4 (7.7%)	
Presence of phenotypic criteria						
Inflammation (*n*, %)	12 (17.9%)	39 (81.3%)	<0.001	21 (31.3%)	31 (59.6%)	0.002
Dietary intake or reduced absorption (*n*, %)	4 (6%)	30 (62.5%)	<0.001	5 (7.5%)	31 (59.6%)	<0.001

Abbreviations: CRP = C-reactive protein; BMI = body mass index; CC = calf circumference; MM = muscle mass; GLIM = Global Leadership Initiative on Malnutrition; SGA = Subjective Global Assessment; mg = milligrams; dL = deciliters; kg = kilograms; m^2^ = meter^2^; cm = centimeters; *n* = number of cases.

**Table 3 nutrients-15-04012-t003:** Concurrent validity of the different study cohorts.

	Whole Cohort	Entire Cohort Excluding Patients with 1 or 2 Nonassessable GLIM Phenotypic Criteria	Entire Cohort Excluding Patients with No Degree of Malnutrition
Sensitivity (%, 95% CI)	78.0 (64.0–88.5)	73.8 (58.0–86.1)	77.6 (63.3–88.2)
Specificity (%, 95% CI)	86.2 (75.3–93.5)	88.7 (77.0–95.7)	98.3 (90.6–99.9)
Positive predictive value (%, 95% CI)	83.6 (75.0–89.7)	83.8 (68.0–93.8)	97.4 (86.5–99.9)
Negative predictive value (%, 95% CI)	81.3 (69.9–89.0)	81.0 (68.6–90.1)	83.6 (72.5–91.5)
AUC ROC (95% CI)	0.82 (0.74–0.90)	0.81 (0.72–0.91)	0.88 (0.81–0.95)
Weighted Kappa (95% CI)	0.64 (0.50–0.79)	0.63 (0.47–0.79)	0.77 (0.65–0.89)

Abbreviations: AUC = area under the curve; CI = confidence interval.

**Table 4 nutrients-15-04012-t004:** Predictive validity of the GLIM criteria and the SGA in relation to hospital stay > 10 days.

	Well Nourished	Malnourished	Raw OR (95% CI)	Adjusted OR (95% CI) Model 1 *	Adjusted OR (95% CI) Model 2 ^†^
GLIM, *n* (%)	21 (43.8%)	12 (17.9%)	3.56 (1.55–8.51)	2.98 (1.21–7.60)	2.96 (1.15–7.69)
SGA, *n* (%)	26 (50%)	8 (11.9%)	7.37 (3.06–19.48)	6.16 (2.42– 7.08)	7.11 (2.60– 19.91)

* Adjusted for sex, age, Charlson index and Barthel index; ^†^ Adjusted for sex, age, Charlson index, and Barthel index, excluding hospital mortality; Abbreviations: OR = Odds ratio; GLIM = Global Leadership Initiative on Malnutrition; SGA = Subjective Global Assessment; *n* = number of cases; CI = Confidence Interval.

**Table 5 nutrients-15-04012-t005:** Patient characteristics by assessment of phenotypic criteria (all or only 1 or 2).

Variables	All Evaluable Criteria (*n* = 95)	1 or 2 Assessable Criteria (*n* = 20)	*p* Value
Sociodemographic data			
Age (years)	63.3 ± 15.1	74.5 ± 10.2	<0.001
Women (%)	47.4	65.0	0.152
Caucasian (%)	96.8	100	0.723
Clinical data			
Charlson index	2.0 ± 2.3	2.2 ± 2.1	0.415
Barthel index	87.2 ± 21.1	64.8 ± 31.5	<0.001
Analytical parameters			
Albumin (mg/dL)	3.0 ± 0.6	3.0 ± 0.4	0.396
CRP (mg/dL)	14.2 ± 34.4	9.3 ± 12.6	0.275
Nutritional characteristics			
Current weight (kg)	71.3 ± 15.9	70.0 ± 9.4	0.416
BMI (kg/m^2^)	26.3 ± 5.9	28.2 ± 3.9	0.294
Weight loss (kg)	2.9 ± 6.0	4.7 ± 8.0	0.237
CC (cm)	32.8 ± 3.5	34.4 ± 6.6	0.076
Days of hospitalization	9.1 ± 9.3	7.6 ± 6.2	0.237

Abbreviations: CRP = C-reactive protein; BMI = body mass index; CC = calf circumference; mg = milligrams; dL = deciliter; kg = kilograms; m^2^ = meter^2^; cm = centimeters.

## Data Availability

No new data were created or analyzed in this study. Data sharing is not applicable to this article.

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
