# Peer review of "Validity and Applicability of the Global Leadership Initiative on Malnutrition (GLIM) Criteria in Patients Hospitalized for Acute Medical Conditions"

_nutrients, 2023, doi:10.3390/nu15184012_

Round 1
Reviewer 1 Report
1. This validation study was conducted in patients hospitalized for acute medical conditions. It may be helpful to provide some background information on the choice of acute medical conditions.
2. Nine out of the 65 patients (13.8%) without malnutrition as determined using the SGA were classified as malnutrition using the GLIM criteria. Considering the clinical importance of proper care for malnutrition, is the benefit risk of SGA over GLIM justifiable?
3. When dichotomizing patients with hospital stay >10 days vs. <=10 days, how were patients who died <10 days handled (if any)?
4. The difference in time spent on completing the GLIM vs SGA does not seem to be meaningful (though statistically significantly different).
Author Response
- This validation study was conducted in patients hospitalized for acute medical conditions. It may be helpful to provide some background information on the choice of acute medical conditions.
In our hospitals, patients were admitted to medical wards, surgical wards, gynecology wards, or orthopedic wards. In the present study, only patients admitted to medical wards for an acute medical issue were included. In accordance with the reviewer's comment, we have enhanced the description of the study's selection criteria.
- Nine out of the 65 patients (13.8%) without malnutrition as determined using the SGA were classified as malnutrition using the GLIM criteria. Considering the clinical importance of proper care for malnutrition, is the benefit risk of SGA over GLIM justifiable?
We appreciate the reviewer's comment. Patients who only exhibit malnutrition according to GLIM criteria should be interpreted as false positives, which favor the superiority of SGA over GLIM. Furthermore, the higher predictive validity of SGA for detecting prolonged hospitalization reinforces SGA as the gold standard.
- When dichotomizing patients with hospital stay >10 days vs. <=10 days, how were patients who died <10 days handled (if any)?
Hospital stay duration was determined without considering mortality. In light of the insightful comment from the reviewer, we have reevaluated the mortality within our cohort. We have identified that 7 patients died during admission, with 4 of them having an admission of less than 10 days. After excluding these patients, the predictive validity of both GLIM and VSG remained significant. We have incorporated all of this information into the revised manuscript.
- The difference in time spent on completing the GLIM vs SGA does not seem to be meaningful (though statistically significantly different).
In accordance with the reviewer's comment, we have provided a more nuanced assessment of the real time-saving impact of SGA compared to GLIM.
Reviewer 2 Report
I’ve read with attention the paper of Fontané et al. that is potentially of interest. The background and aim of the study have been clearly defined. The methodology applied is overall correct, the results are reliable and adequately discussed. I’ve only some minor comments:
- The acronyms have to be spelled when firstly mentioned in the paper (including title and abstract)
- The discussion section is very long and a bit self-congratulatory. Among study limitation, the authors should also stress that their patient sample was small and their observation limited to a single center, where it could not directly extended to any other settings.
- A graphical abstract could more easily attract the interest of the readers toward the paper.
- As a minor comment, it is not clear why so many authors were needed to carry out a relatively small and easy study
Author Response
I’ve read with attention the paper of Fontané et al. that is potentially of interest. The background and aim of the study have been clearly defined. The methodology applied is overall correct, the results are reliable and adequately discussed. I’ve only some minor comments:
- The acronyms have to be spelled when firstly mentioned in the paper (including title and abstract)
In accordance with the reviewer's comment, we have included the acronyms the first time they appear in the text.
- The discussion section is very long and a bit self-congratulatory. Among study limitation, the authors should also stress that their patient sample was small and their observation limited to a single center, where it could not directly extended to any other settings.
In accordance with the reviewer's comments, we have clarified the study's limitations.
- A graphical abstract could more easily attract the interest of the readers toward the paper.
To our knowledge, Nutrients does not accept graphical abstracts.
- As a minor comment, it is not clear why so many authors were needed to carry out a relatively small and easy study
Author contributions are described in the manuscript. As the corresponding author and leader of this study, all authors meet sufficient criteria to be listed as authors of the article.